# CONDITIONAL GENERATIVE MODELING VIA LEARNING THE LATENT SPACE

**Sameera Ramasinghe**[*†⋆], **Kanchana Ranasinghe**[‡], **Salman Khan**[‡†],
**Nick Barnes**[†] **and Stephen Gould**[†]
[†]Australian National University, [⋆]Data61 (CSIRO), [‡]Mohamed bin Zayed University of AI
sameera.ramasinghe@anu.edu.au

## ABSTRACT

Although deep learning has achieved appealing results on several machine learning tasks, most of the models are deterministic at inference, limiting their application to single-modal settings. We propose a novel general-purpose framework for conditional generation in multimodal spaces, that uses latent variables to model generalizable learning patterns while minimizing a family of regression cost functions. At inference, the latent variables are optimized to find solutions corresponding to multiple output modes. Compared to existing generative solutions, our approach demonstrates faster and more stable convergence, and can learn better representations for downstream tasks. Importantly, it provides a simple generic model that can perform better than highly engineered pipelines tailored using domain expertise on a variety of tasks, while generating diverse outputs. Code available at https://github.com/samgregoost/cGML.

## 1 INTRODUCTION

Conditional generative models provide a natural mechanism to jointly learn a data distribution and optimize predictions. In contrast, discriminative models improve predictions by modeling the label distribution. Learning to model the data distribution allows generating novel samples and is considered a preferred way to understand the real world. Existing conditional generative models have generally been explored in single-modal settings, where a one-to-one mapping between input and output domains exists (Nalisnick et al., 2019; Fetaya et al., 2020). Here, we investigate continuous multimodal (CMM) spaces for generative modeling, where one-to-many mappings exist between input and output domains. This is critical since many real world situations are inherently multimodal, e.g., humans can imagine several completions for a given occluded image. In a discrete setting, this problem becomes relatively easy to tackle using techniques such as maximum-likelihood-estimation, since the output can be predicted as a vector (Zhang et al., 2016), which is not possible in continuous domains. One way to model CMM spaces is by using variational inference, e.g., variational autoencoders (VAE) (Kingma & Welling, 2013). However, the approximated posterior distribution of VAEs are often restricted to the Gaussian family, which hinders its ability to model more complex distributions. As a solution, Maaløe et al. (2016) suggested using auxiliary variables to improve the variational distribution. To this end, the latent variables are hierarchically correlated through injected auxiliary variables, which can produce non-Gaussian distributions. A slightly similar work by Rezende & Mohamed (2015) proposed Normalizing Flows, that can hierarchically generate more complex probability distributions by applying a series of bijective mappings to an original simpler distribution. Recently, Chang et al. (2019) proposed a model, where a separate variable can be used to vary the impact of different loss components at inference, which allows diverse outputs. For a more detailed discussion on these methods see App. 1.

In addition to the aforesaid methods, in order to model CMM spaces, a prominent approach in the literature is to use a combination of reconstruction and adversarial losses (Isola et al., 2017; Zhang et al., 2016; Pathak et al., 2016). However, this entails key shortcomings. 1) The goals of adversarial and reconstruction losses are contradictory (Sec. 4), hence model engineering and numerous regularizers are required to support convergence (Lee et al., 2019; Mao et al., 2019),

---

[*]Corresponding author.

thereby resulting in less-generic models tailored for specific applications (Zeng et al., 2019; Vitoria et al., 2020). 2) The adversarial loss based models are notorious for difficult convergence due to the challenge of finding Nash equilibrium of a non-convex min-max game in high-dimensions (Barnett, 2018; Chu et al., 2020; Kodali et al., 2017). 3) The convergence is heavily dependent on the architecture, hence such models show lack of scalability (Thanh-Tung et al., 2019; Arora & Zhang, 2017). 4) The promise of assisting downstream tasks remains challenging, with a large gap in performance between the generative modelling approaches and their discriminative counterparts (Grathwohl et al., 2020; Jing & Tian, 2020).

In this work, we propose a general-purpose framework—Conditional Generation by Modeling the Latent Space (cGML)—for modeling CMM spaces using a set of domain-agnostic regression cost functions instead of the adversarial loss. This improves both the stability and eliminates the incompatibility between the adversarial and reconstruction losses, allowing more precise outputs while maintaining diversity. The underlying notion is to learn the '*behaviour of the latent variables*' in minimizing these cost functions while converging to an optimum mode during the training phase, and mimicking the same at inference. Despite being a novel direction, the proposed framework showcases promising attributes by: (a) achieving state-of-the-art results on a diverse set of tasks using a generic model, implying generalizability, (b) rapid convergence to optimal modes despite architectural changes, (c) learning useful features for downstream tasks, and (d) producing diverse outputs via traversal through multiple output modes at inference.

## 2 PROPOSED METHODOLOGY

We define a family of cost functions $\{E_{i,j} = d(y^g_{i,j}, \mathcal{G}(x_j, w))\}$, where $x_j \sim \chi$ is the input, $y^g_{i,j} \sim \Upsilon$ is the $i^{th}$ ground-truth mode for $x_j$, $\mathcal{G}$ is a generator function with weights $w$, and $d(\cdot, \cdot)$ is a distance function. Note that the number of cost functions $E_{(\cdot,j)}$ for a given $x_j$ can vary over $\chi$. Our aim here is to come up with a generator function $\mathcal{G}(x_j, w)$, that can minimize each $E_{i,j}, \forall i$ as $\mathcal{G}(x_j, w) \rightarrow y^g_{i,j}$. However, since $\mathcal{G}$ is a deterministic function ($x$ and $w$ are both fixed at inference), it can only produce a single output. Therefore, we introduce a latent vector $z$ to the generator function, that can be used to converge $\bar{y}_{i,j} = \mathcal{G}(x_j, w, z_{i,j})$ towards a ground truth $y^g_{(i,j)}$ at inference, and possibly, to multiple solutions. Formally, the family of cost functions now becomes: $\{\hat{E}_{i,j} = d(y^g_{i,j}, \mathcal{G}(x_j, w, z_{i,j}))\}, \forall z_{i,j} \sim \zeta$. Then, our training objective can be defined as finding a set of optimal $z_i^* \in \zeta$ and $w^* \in \omega$ by minimizing $\mathbb{E}_{i \sim I}[\hat{E}_{i,j}]$, where $I$ is the number of possible solutions for $x_j$. Note that $w^*$ is fixed for all $i$ and a different $z_i^*$ exists for each $i$. Considering all the training samples $x_j \sim \chi$, our training objective becomes,

$$\{\{z^*_{i,j}\}, w^*\} = \underset{z_{i,j} \in \zeta, w \in \omega}{\arg\min} \ \mathbb{E}_{i \in I, j \in J}[\hat{E}_{i,j}]. \tag{1}$$

Eq. 1 can be optimized via Algorithm 1 (proof in App. 2.2). Intuitively, the goal of Eq. 1 is to obtain a family of optimal latent codes $\{z^*_{i,j}\}$, each causing a global minima in the corresponding $\hat{E}_{i,j}$ as $y^g_{i,j} = \mathcal{G}(x_j, w, z^*_{i,j})$. Consequently, at inference, we can optimize $\bar{y}_{i,j}$ to converge to an optimal mode in the output space by varying $z$. Therefore, we predict an estimated $\bar{z}_{i,j}$ at inference,

$$\bar{z}_{i,j} \approx \min_z \hat{E}_{i,j}, \tag{2}$$

for each $y^g_{i,j}$, which in turn can be used to obtain the prediction $\mathcal{G}(x_j, w, \bar{z}_{i,j}) \approx y^g_{i,j}$. In other words, for a selected $x_j$, let $\bar{y}^t_{i,j}$ be the initial estimate for $\bar{y}_{i,j}$. At inference, $z$ can traverse gradually towards an optimum point $y^g_{i,j}$ in the space, forcing $\bar{y}^{t+n}_{i,j} \rightarrow y^g_{i,j}$, in finite steps ($n$).

However, still a critical problem exists: Eq. 2 depends on $y^g_{i,j}$, which is not available at inference. As a remedy, we enforce Lipschitz constraints on $\mathcal{G}$ over $(x_j, z_{i,j})$, which bounds the gradient norm as,

$$\frac{\left\| \mathcal{G}(x_j, w^*, z^*_{i,j}) - \mathcal{G}(x_j, w^*, z_0) \right\|}{\left\| z^*_{i,j} - z_0 \right\|} \leq \int \left\| \nabla_z \mathcal{G}(x_j, w^*, \gamma(t)) \right\| dt \leq C, \tag{3}$$

where $z_0 \sim \zeta$ is an arbitrary random initialization, $C$ is a constant, and $\gamma(\cdot)$ is a straight path from $z_0$ to $z^*_{i,j}$ (proof in App. 2.1) . Intuitively, Eq. 3 implies that the gradients $\nabla_z \mathcal{G}(x_j, w^*, z_0)$ along the path $\gamma(\cdot)$ do not tend to vanish or explode, hence, finding the path to optimal $z^*_{i,j}$ in the space $\zeta$ becomes a fairly straight forward regression problem. Moreover, enforcing the Lipschitz constraint

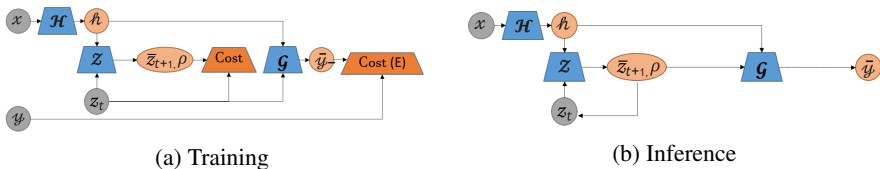

(a) Training                    (b) Inference

Figure 1: Training and inference process. Refer to Algorithm 1 for the training process. At inference, $z$ is iteratively updated using the predictions of $\mathcal{Z}$ and fed to $\mathcal{G}$ to obtain increasingly fine-tuned outputs (see Sec. 3).

encourages meaningful structuring of the latent space: suppose $z^*_{1,j}$ and $z^*_{2,j}$ are two optimal codes corresponding to two ground truth modes for a particular input. Since $\|z^*_{2,j} - z^*_{1,j}\|$ is lower bounded by $\frac{\left\|\mathcal{G}(x_j, w^*, z^*_{2,j}) - \mathcal{G}(x_j, w^*, z^*_{1,j})\right\|}{L}$, where $L$ is the Lipschitz constant, the minimum distance between the two latent codes is proportional to the difference between the corresponding ground truth modes. In practice, we observed that this encourages the optimum latent codes to be placed sparsely (visual illustration in App. 2), which helps a network to learn distinctive paths towards different modes.

## 2.1 CONVERGENCE AT INFERENCE

We formulate finding the convergence path of $z$ at inference as a regression problem, i.e., $z_{t+1} = r(z_t, x_j)$. We implement $r(\cdot)$ as a recurrent neural network (RNN). The series of predicted values $\{z_{(t+k)} : k = 1, 2, .., N\}$ can be modeled as a first-order Markov chain requiring no memory for the RNN. We observe that enforcing Lipschitz continuity on $\mathcal{G}$ over $z$ leads to smooth trajectories even in high dimensional settings, hence, memorizing more than one step into the history is redundant. However, $z_{t+1}$ is not a state variable, i.e., the existence of multiple modes for output prediction $\bar{y}$ leads to multiple possible solutions for $z_{t+1}$. On the contrary, $\mathbb{E}[z_{t+1}]$ is a state variable w.r.t. the state $(z_t, x)$, which can be used as an approximation to reach the optimal $z^*$ at inference. Therefore, instead of directly learning $r(\cdot)$, we learn a simplified version $r'(z_t, x) = \mathbb{E}[z_{t+1}]$. Intuitively, the whole process can be understood as observing the behavior of $z$ on a smooth surface at the training stage, and predicting the movement at inference. A key aspect of $r'(z_t, x)$ is that the model is capable of converging to multiple possible optimum modes at inference based on the initial position of $z$.

## 2.2 MOMENTUM AS A SUPPLEMENTARY AID

Based on Sec. 2.1, $z$ can now traverse to an optimal position $z^*$ during inference. However, there can exist rare symmetrical positions in the $\zeta$ where $\mathbb{E}[z_{t+1}] - z_t \approx 0$, although far away from $\{z^*\}$, forcing $z_{t+1} \approx z_t$. Simply, the above phenomenon can occur if some $z_{t+1}$ has traveled in many non-orthogonal directions, so the vector addition of $z_{t+1} \approx 0$. This can *fool* the system to falsely identify convergence points, forming *phantom* optimum point distributions amongst the true distribution (see Fig. 3). To avoid such behavior, we learn the expected momentum $\mathbb{E}[\rho(z_t, x_j)] = \alpha \mathbb{E}[|z_{t+1} - z_t|x_j]$ at each $(z_t, x_j)$ during the training phase, where $\alpha$ is an empirically chosen scalar. In practice, $\mathbb{E}[\rho(z_t, x_j)] \to 0$ as $z_{t+1}, z_t \to \{z^*\}$. Thus, to avoid *phantom* distributions, we improve the $z$ update as,

$$z_{t+1} = z_t + \mathbb{E}[\rho(z_t, x_j)] \left[ \frac{r'(z_t, x_j) - z_t}{\|r'(z_t, x_j) - z_t\|} \right]. \tag{4}$$

Since both $\mathbb{E}[\rho(z_t, x_j)]$ and $r'(z_t, x_j)$ are functions on $(z_t, x_j)$, we jointly learn these two functions using a single network $\mathcal{Z}(z_t, x_j)$. Note that coefficient $\mathbb{E}[\rho(z_t, x_j)]$ serves two practical purposes: 1) slows down the movement of $z$ near true distributions, 2) pushes $z$ out of the phantom distributions.

## 3 OVERALL DESIGN

The proposed model consists of three major blocks as shown in Fig. 1: an encoder $\mathcal{H}$, a generator $\mathcal{G}$, and $\mathcal{Z}$. The detailed architecture diagram for $128 \times 128$ is shown in Fig. 2. Note that for derivations in Sec. 2, we used $x$ instead of $h = \mathcal{H}(x)$, as $h$ is a high-level representation of $x$. The training process is illustrated in Algorithm 1. At each optimization $z_{t+1} = z_t - \beta \nabla_{z_t}[\hat{E}_{i,j}]$, $\mathcal{Z}$ is trained separately to approximate $(z_{t+1}, \rho)$. At inference, $x$ is fed to $\mathcal{H}$, and then $\mathcal{Z}$ optimizes the output $\bar{y}$ by updating $z$ for a pre-defined number of iterations of Eq. 4. For $\hat{E}(\cdot, \cdot)$, we use $L_1$ loss. Furthermore, it is important to limit the search space for $z_{t+1}$, to improve the performance of $\mathcal{Z}$. To this end, we

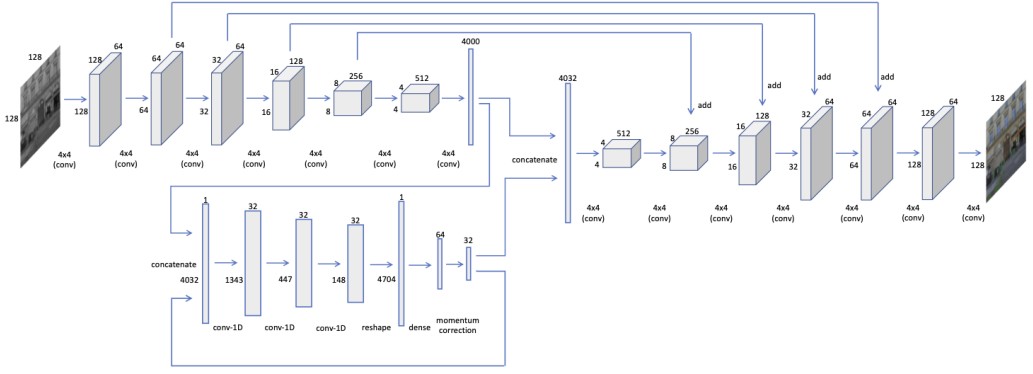

Figure 2: *Overall architecture for $128 \times 128$ inputs.*

---

**Algorithm 1:** Training algorithm

---
sample inputs $\{x_1, x_2, ..., x_J\} \in \chi$; sample outputs $\{y_1, y_2, ..., y_J\} \in \Upsilon$ ;
**for** $k$ *epochs* **do**
    **for** $x$ *in* $\chi$ **do**
        **for** $l$ *steps* **do**
            update $z = \{z_1, z_2, ..., z_J\}$: $\nabla_z \hat{E}$                    ▷ Freeze $\mathcal{H}, \mathcal{G}, \mathcal{Z}$ and update $z$
            update $\mathcal{Z}$: $\nabla_w L_1[(z_{t+1}, \rho), \mathcal{Z}(z_t, \mathcal{H}(x))]$          ▷ Freeze $\mathcal{H}, \mathcal{G}, z$ and update $\mathcal{Z}$
        update $\mathcal{G}, \mathcal{H}$: $\nabla_w \hat{E}$                             ▷ Freeze $\mathcal{Z}, z$ and update $\mathcal{H}, \mathcal{G}$

---

sample $z$ from the surface of the $n$-dimensional sphere ($\mathbb{S}^n$). Moreover, to ensure faster convergence of the model, we force Lipschitz continuity on both $\mathcal{Z}$ and the $\mathcal{G}$ (App. 2.4). For hyper-parameters and training details, see App. 3.1.

## 4 MOTIVATION

Here, we explain the drawbacks of conditional GAN methods and illustrate our idea via a toy example.

**Incompatibility of adversarial and reconstruction losses:** cGANs use a combination of adversarial and reconstruction losses. We note that this combination is suboptimal to model CMM spaces.
***Remark:*** *Consider a generator $G(x, z)$ and a discriminator $D(x, z)$, where $x$ and $z$ are the input and the noise vector, respectively. Then, consider an arbitrary input $x_j$ and the corresponding set of ground-truths $\{y_{i,j}^g\}, i = 1, 2, ..N$. Further, let us define the optimal generator $G^*(x_j, z) = \hat{y}, \hat{y} \in \{y_{i,j}^g\}$, $L_{GAN} = \mathbb{E}_i[\log D(y_{i,j}^g)] + \mathbb{E}_z[\log(1 - D(G(x_j, z))]$ and $L_\ell = \mathbb{E}_{i,z}[|y_{i,j}^g - G(x_j, z)|]$. Then, $G^* \neq \hat{G}^*$ where $\hat{G}^* = \arg \min_G \max_D L_{GAN} + \lambda L_\ell, \forall \lambda \neq 0$. (Proof in App. 2.3).*

**Generalizability:** The incompatibility of above mentioned loss functions demands domain specific design choices from models that target high realism in CMM settings. This hinders the generalizability across different tasks (Vitoria et al., 2020; Zeng et al., 2019). We further argue that due to this discrepancy, cGANs learn sub-optimal features which are less useful for downstream tasks (Sec. 5.3).

**Convergence and the sensitivity to the architecture:** The difficulty of converging GANs to the Nash equilibrium of a non-convex min-max game in high-dimensional spaces is well explored (Barnett, 2018; Chu et al., 2020; Kodali et al., 2017). Goodfellow et al. (2014b) underlines *if the discriminator has enough capacity, and is optimal at every step of the GAN algorithm, then the generated distribution converges to the real distribution*; that cannot be guaranteed in a practical scenario. In fact, Arora et al. (2018) confirmed that the adversarial objective can easily approach to an equilibrium even if the generated distribution has very low support, and further, the number of training samples required to avoid mode collapse can be in order of $\exp(d)$ ($d$ is the data dimension).

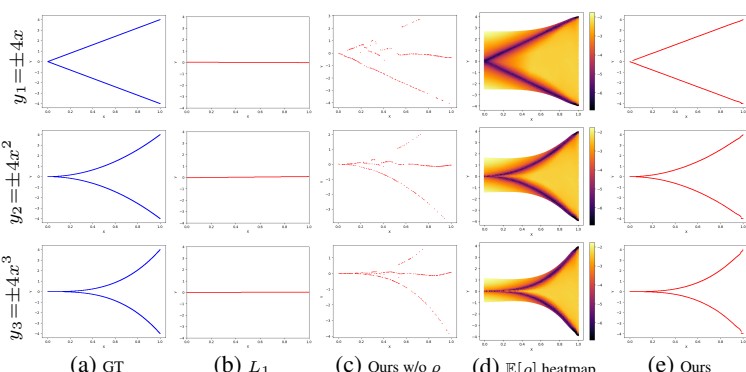

Figure 3: *Toy Example:* Plots generated for each dimension of the CMM space $\Upsilon$. (a) Ground-truth distributions. (b) Model outputs for $L_1$ loss. (c) Output when trained with the proposed objective (without $\rho$ correction). Note the *phantom distribution* identified by the model. (d) $\mathbb{E}[\rho]$ as a heatmap on $(x, y)$. $\mathbb{E}[\rho]$ is lower near the true distribution and higher otherwise. (e) Model outputs after $\rho$ correction.

(a) GT    (b) $L_1$    (c) Ours w/o $\rho$    (d) $\mathbb{E}[\rho]$ heatmap    (e) Ours

**Multimodality:** The ability to generate diverse outputs, i.e., convergence to multiple modes in the output space, is an important requirement. Despite the typical noise input, cGANs generally lack the ability to generate diverse outputs (Lee et al., 2019). Pathak et al. (2016) and Iizuka et al. (2016) even state that better results are obtained when the noise is completely removed. Further, variants of cGAN that target diversity often face a trafe-off between the realism and diversity (He et al., 2018), as they have to compromise between the reconstruction and adversarial losses.

**A toy example:** Here, we experiment with the formulations in Sec. 2. Consider a 3D CMM space $y = \pm 4(x, x^2, x^3)$. Then, we construct multi-layer perceptrons (MLP) with three layers to represent each of the functions, $\mathcal{H}$, $\mathcal{G}$, and $\mathcal{Z}$, and compare the proposed method against the $L_1$ loss. Figure 3 illustrates the results. As expected, $L_1$ loss generates the line $y = 0$, and is inadequate to model the multimodal space. As explained in Sec. 2.2, without momentum correction, the network is fooled by a phantom distribution where $\mathbb{E}[z_{t+1}] \approx 0$ at training time. However, the *push* of momentum removes the phantom distribution and refines the output to closely resemble the input distribution. As implied in Sec. 2.2, the momentum is maximized near the true distribution and minimized otherwise.

## 5 EXPERIMENTS AND DISCUSSIONS

The distribution of natural images lies on a high dimensional manifold, making the task of modelling it extremely challenging. Moreover, conditional image generation poses an additional challenge with their constrained multimodal output space (a single input may correspond to multiple outputs while not all of them are available for training). In this section, we experiment on several such tasks. For a fair comparison with a similar capacity GAN, we use the encoder and decoder architectures used in Pathak et al. (2016) for $\mathcal{H}$ and $\mathcal{G}$ respectively. We make two minor modifications: the channel-wise fully connected (FC) layers are removed and U-Net style skip connections are added (see App. 3.1). We train the existing models for a maximum of 200 epochs where pretrained weights are not provided, and demonstrate the generalizability of our theoretical framework in diverse practical settings by using a generic network for all the experiments. Models used for comparisons are denoted as follows: PN (Zeng et al., 2019), CA (Yu et al., 2018b), DSGAN (Yang et al., 2019), CIC (Zhang et al., 2016), RFR (Li et al., 2020), Chroma (Vitoria et al., 2020), P2P (Isola et al., 2017), Iizuka (Iizuka et al., 2016), CE (Pathak et al., 2016), CRN (Chen & Koltun, 2017a), and B-GAN (Zhu et al., 2017b).

### 5.1 CORRUPTED IMAGE RECOVERY

We design this task as image completion, i.e., given a masked image as input, our goal is to recover the masked area. Interestingly, we observed that the MNIST dataset, in its original form, does not have a multimodal behaviour, i.e., a fraction of the input image only maps to a single output. Therefore, we modify the training data as follows: first, we overlap the top half of an input image with the top half of another randomly sampled image. We carry out this corruption for 20% of the training data. Corrupted samples are not fixed across epochs. Then, we apply a random sized mask to the top half, and ask the network to predict the missing pixels. We choose two competitive baselines here: our network with the $L_1$ loss and CE. Fig. 4 illustrates the predictions. As shown, our model converges to the most probable non-corrupted mode without any ambiguity, while other baselines give sub-optimal results. In the next experiment, we add a small white box to the top part of the ground-truth images at

| Method | User study | | Turing test |
|---|---|---|---|
| | STL | ImageNet | ImageNet |
| Iizuka et al. | 21.89 | 32.28 | - |
| Chroma | 32.40 | 31.67 | - |
| Ours | **45.71** | **36.05** | 31.66 |

Table 1: *Colorization:* Psychophysical study and Turing test results. All performances are in %.

| Method | STL | | | | ImageNet | | | |
|---|---|---|---|---|---|---|---|---|
| | LPIP ↓ | PieAPP ↓ | SSIM ↑ | PSNR ↑ | LPIP ↓ | PieAPP ↓ | SSIM ↑ | PSNR ↑ |
| Iizuka et al. | 0.18 | 2.37 | 0.81 | 24.30 | 0.17 | 2.47 | 0.87 | 18.43 |
| P2P | 1.21 | 2.69 | 0.73 | 17.80 | 2.01 | 2.80 | 0.87 | 18.43 |
| CIC | 0.18 | 2.81 | 0.71 | 22.04 | 0.19 | 2.56 | 0.71 | 19.11 |
| Chroma | 0.16 | 2.06 | 0.91 | 25.57 | **0.16** | 2.13 | 0.90 | 23.33 |
| Ours | **0.12** | **1.47** | **0.95** | **27.03** | 0.16 | 2.04 | **0.92** | **24.51** |
| Ours (w/o ρ) | 0.16 | 1.90 | 0.89 | 25.02 | 0.20 | 2.11 | 0.88 | 23.21 |

Table 2: *Colorization:* Quantitative analysis of our method against the state-of-the-art. Ours perform better on a variety of metrics.

| Method | 10% corruption | | | | 15% corruption | | | | 25% corruption | | | |
|---|---|---|---|---|---|---|---|---|---|---|---|---|
| | LPIP ↓ | PieAPP ↓ | PSNR ↑ | SSIM ↑ | LPIP ↓ | PieAPP ↓ | PSNR ↑ | SSIM ↑ | LPIP ↓ | PieAPP ↓ | PSNR ↑ | SSIM ↑ |
| DSGAN | 0.101 | 1.577 | 20.13 | 0.67 | 0.189 | 2.970 | 18.45 | 0.55 | 0.213 | 3.54 | 16.44 | 0.49 |
| PN | **0.045** | **0.639** | 27.11 | 0.88 | 0.084 | **0.680** | 20.50 | 0.71 | 0.147 | 0.764 | 19.41 | 0.63 |
| CE | 0.092 | 1.134 | 22.34 | 0.71 | 0.134 | 2.134 | 19.11 | 0.63 | 0.189 | 2.717 | 17.44 | 0.51 |
| P2P | 0.074 | 0.942 | 22.33 | 0.79 | 0.101 | 1.971 | 19.34 | 0.70 | 0.185 | 2.378 | 17.81 | 0.57 |
| CA | 0.048 | 0.731 | 26.45 | 0.83 | 0.091 | 0.933 | 20.12 | 0.72 | 0.166 | 0.822 | 21.43 | 0.72 |
| RFR | 0.051 | 0.743 | **29.31** | 0.85 | 0.097 | 1.033 | 19.22 | 0.70 | 0.171 | 1.127 | 18.42 | 0.61 |
| Ours (w/o ρ) | 0.053 | 0.799 | 27.77 | 0.83 | 0.085 | 0.844 | 23.22 | 0.76 | 0.141 | 0.812 | 22.31 | 0.74 |
| Ours | 0.051 | 0.727 | 27.83 | **0.89** | **0.080** | 0.740 | **26.43** | **0.80** | **0.129** | **0.760** | **24.16** | **0.77** |

Table 3: *Image completion:* Quantitative analysis of our method against state-of-the-art on a variety of metrics.

different rates. At inference, our model was able to converge to both the modes (Fig. 5), depending on the initial position of $z$, as the probability of the alternate mode reaches 0.3.

## 5.2 AUTOMATIC IMAGE COLORIZATION

Deep models have tackled this problem using semantic priors (Iizuka et al., 2016; Vitoria et al., 2020), adversarial and $L_1$ losses (Isola et al., 2017; Zhu et al., 2017a; Lee et al., 2019), or by conversion to a discrete form through binning of color values (Zhang et al., 2016). Although these methods provide compelling results, several inherent limitations exist: (a) use of semantic priors results in complex models, (b) adversarial loss suffers from drawbacks (see Sec. 4), and (c) discretization reduces the precision. In contrast, we achieve better results using a simpler model.

The input and the output of the network are $l$ and $(a, b)$ planes respectively (LAB color space). However, since the color distributions of $a$ and $b$ spaces are highly imbalanced over a natural dataset (Zhang et al., 2016), we add another constraint to the cost function $E$ to push the predicted $a$ and $b$ colors towards a uniform distribution: $E = \|a_{gt} - a\| + \|b_{gt} - b\| + \lambda(loss_{kl,a} + loss_{kl,b})$, where $loss_{kl,\cdot} = \mathrm{KL}(\cdot\|u(0,1))$. Here, $\mathrm{KL}(\cdot\|\cdot)$ is the KL divergence and $u(0,1)$ is a uniform distribution (see App. 3.3). Fig. 7 and Table 2 depict our qualitative and quantitative results, respectively. We demonstrate the superior performance of our method against four metrics: LPIP, PieAPP, SSIM and PSNR (App. 3.2). Fig. 10 depicts examples of multimodality captured by our model (more examples in App. 3.4). Fig. 6 shows colorization behaviour as the $z$ converges during inference.

**User study:** We also conduct two user studies to further validate the quality of generated samples (Table 1). **a)** In the PSYCHOPHYSICAL STUDY, we present volunteers with batches of 3 images, each generated with a different method. A batch is displayed for 5 secs and the user has to pick the most realistic image. After 5 secs, the next image batch is displayed. **b)** We conduct a TURING TEST to validate our output quality against the ground-truth, following the setting proposed by Zhang et al. (2016). The volunteers are presented with a series of paired images (ground-truth and our output). The images are visible for 1 sec, and then the user has an unlimited time to pick the realistic image.

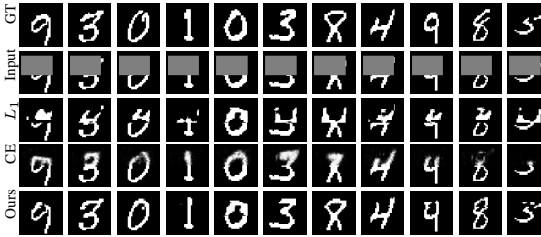

Figure 4: Performance with 20% corrupted data. Our model demonstrates better convergence compared to $L_1$ loss and a similar capacity GAN (Pathak et al., 2016).

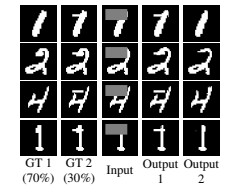

Figure 5: With >30% alternate mode data, our model can converge to both the input modes (cols 4-5).

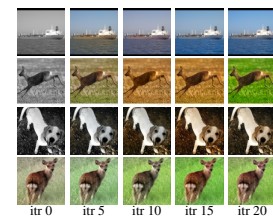

Figure 6: The prediction quality increases as the $z$ traverses to an optimum position at the inference.

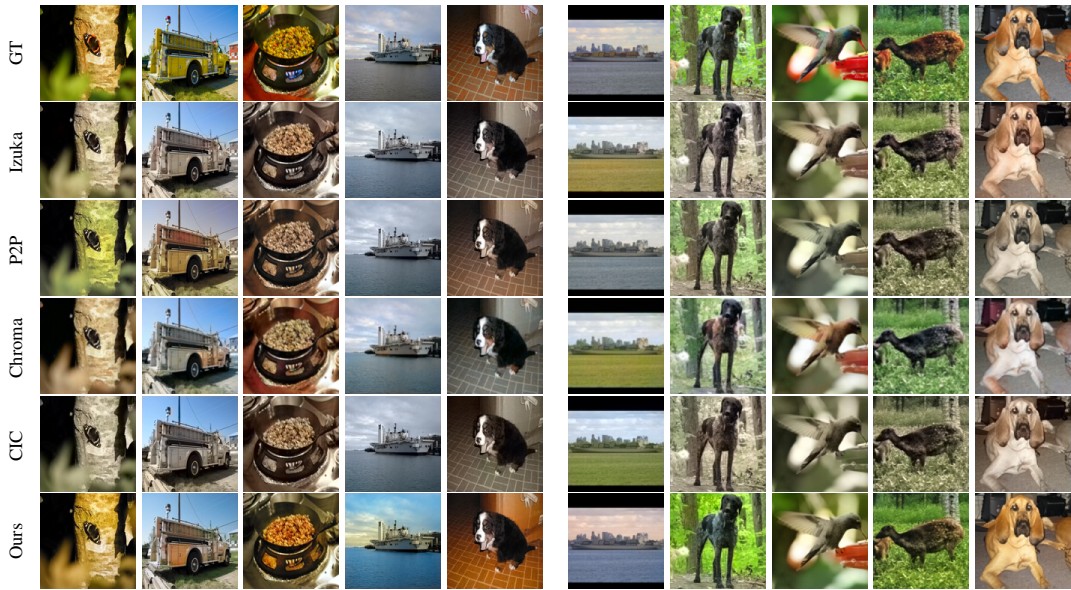

Figure 7: Qualitative comparison against the state-of-the-art on ImageNet (left 5 columns) and STL (right 5 columns) datasets. Our model generally produces more vibrant and balanced color distributions.

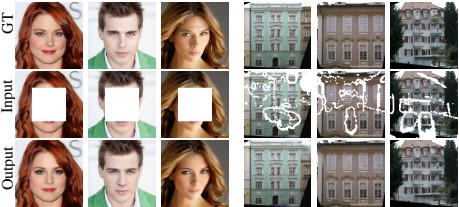

Figure 8: Image completion on Celeb-HQ (left) and Facade (right) datasets. We used fixed center masks and random irregular masks (Liu et al., 2018) for Celeb-HQ and Facades datasets, respectively.

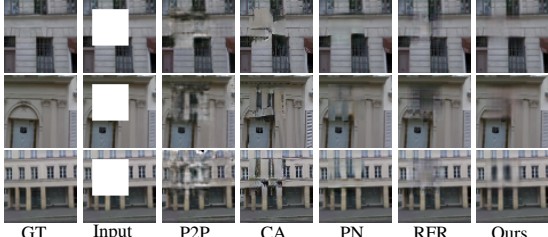

Figure 9: Qualitative comparison for image completion with 25% missing data (models trained with random sized square masks).

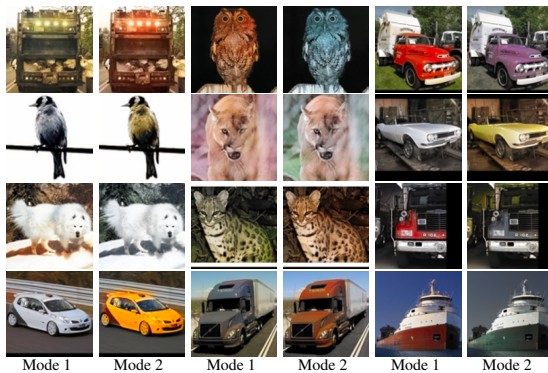

Figure 10: Multiple colorization modes predicted by our model for a single input. *(Best viewed in color)*.

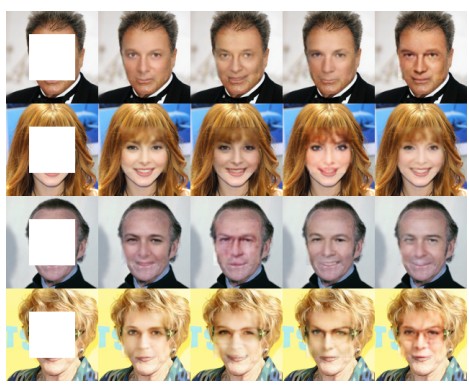

Figure 11: Multi-modality of our predictions on Celeb-HQ dataset. *(Best viewed with zoom)*

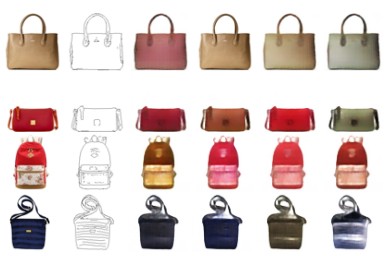

Figure 12: Translation from hand-bag sketches to images.

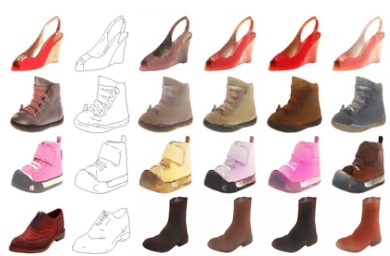

Figure 13: Translation from shoe sketches to images.

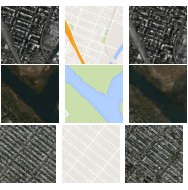

Figure 14: Map to aerial image translation. *From left*: GT, Input and Output. Also see App. 5.2.

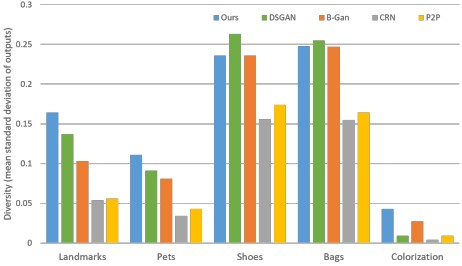

Figure 15: Diversity: Quantitative comparisons.

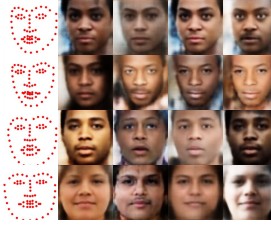

Figure 16: Translation from facial landmarks to faces.

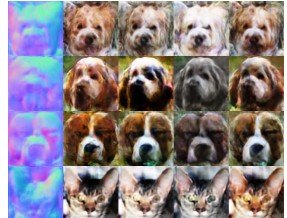

Figure 17: Translation from surface-normals to pet faces.

## 5.3 IMAGE COMPLETION

In this case, we show that our generic model outperforms a similar capacity GAN (CE) as well as task-specific GANs. In contrast to task-specific models, we do not use any domain-specific modifications to make our outputs perceptually pleasing. We observe that with random irregular and fixed-sized masks, all the models perform well, and we were not able to visually observe a considerable difference (Fig. 8, see App. 3.11 for more results). Therefore, we presented models with a more challenging task: train with random sized square-shaped masks and evaluate the performance against masks of varying sizes. Fig. 9 illustrates qualitative results of the models with 25% masked data. As evident, our model recovers details more accurately compared to the state-of-the-art. Notably, all models produce comparable results when trained with a fixed sized center mask, but find this setting more challenging. Table 3 includes a quantitative comparison. Observe that in the case of smaller sized masks, PN performs slightly better than ours, but worse otherwise. We also evaluate the learned features of the models against a downstream classification task (Table 5). First, we train all the models on Facades (Tyleček & Šára, 2013) against random masks, and then apply the trained models on CIFAR10 (Krizhevsky et al., 2009) to extract bottleneck features, and finally pass them through a FC layer for classification (App. 3.7). We compare PN and ours against an oracle (AlexNet features pre-trained on ImageNet) and show our model performs closer to the oracle.

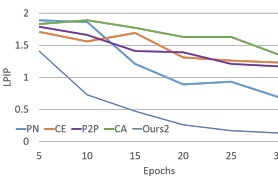

Figure 18: Convergence on image completion (Paris view). Our model exhibits rapid and stable convergence compared to state-of-the-art (PN, CE, P2P, CA).

| Method | M10 | M40 |
|---|---|---|
| Sharma et al. (2016) | 80.5% | 75.5% |
| Han et al. (2019) | 92.2% | 90.2% |
| Achlioptas et al. (2017) | **95.3%** | 85.7% |
| Yang et al. (2018) | 94.4% | 88.4% |
| Sauder & Sievers (2019) | 94.5% | 90.6% |
| Ramasinghe et al. (2019c) | 93.1% | - |
| Khan et al. (2019) | 92.2% | - |
| Ours | 92.4% | **90.9%** |

Table 4: Downstream 3D object classification results on ModelNet10 and ModelNet40 using features learned in an unsupervised manner. All results in % accuracy.

| Method | Pretext | Acc. (%) |
|---|---|---|
| ResNet* | ImageNet Cls. | 74.2 |
| PN | Im. Completion | 40.3 |
| Ours | Im. Completion | **62.5** |

Table 5: Comparison on downstream task (CIFAR10 cls). (*) denotes the oracle case.

| Method | M10 | M40 |
|---|---|---|
| CE | 10.3 | 4.6 |
| cVAE | 8.7 | 4.2 |
| Ours | **84.2** | **79.4** |

Table 6: Reconstruction mAP of 3d spectral denoising.

| Model | CE | PN | Chroma | CIC | P2P | Iizuka et al. | RFR | Ours |
|---|---|---|---|---|---|---|---|---|
| **FLOPS** ($1 \times 10^9$) | 0.634 | 0.946 | 1.275 | 52.839 | 0.732 | 14.082 | 25.64 | 0.638 |

Table 7: Model complexity comparison.

### 5.3.1 DIVERSITY AND OTHER COMPELLING ATTRIBUTES

We also experiment on a diverse set of image translation tasks to demonstrate our generalizability. Fig. 12, 13, 14, 16 and 17 illustrate the qualitative results of *sketch-to-handbag*, *sketch-to-shoes*, *map-to-arial*, *lanmarks-to-faces* and *surface-normals-to-pets* tasks. Fig. 10, 11, 12, 13, 16 and 17 show the ability of our model to converge to multiple modes, depending on the $z$ initialization. Fig. 15 demonstrates the quantitative comparison against other models. See App. 3.4 for further details on experiments. Another appealing feature of our model is its strong convergence properties irrespective of the architecture, hence, *scalability* to different input sizes. Fig. 19 shows examples from image completion and colorization for varying input sizes. We add layers to the architecture to be trained on increasingly high-resolution inputs, where our model was able to converge to optimal modes at each scale (App. 3.8). Fig. 18 demonstrates our faster and stable *convergence*. Table 7 compares the number of FLOPS required by the models for a batch size of 10.

### 5.4 DENOISING OF 3D OBJECTS IN SPECTRAL SPACE

Spectral moments of 3D objects provide a compact representation, and help building light-weight networks (Ramasinghe et al., 2020; 2019b; Cohen et al., 2018; Esteves et al., 2018). However, spectral information of 3D objects has not been used before for self-supervised learning, a key reason being the difficulty of learning representations in the spectral domain due to the complex structure and unbounded spectral coefficients. Here, we present an efficient pretext task that is conducted in the spectral domain: denoising 3D spectral maps. We use two types of spectral spaces: spherical harmonics and Zernike polynomials (App. 4). We first convert the 3D point clouds to spherical harmonic coefficients, arrange the values as a 2D map, and mask or add noise to a map portion (App. 3.12). The goal is to recover the original spectral map. Fig. 20 and Table 6 depicts our qualitative and quantitative results. We perform favorably well against other methods. To evaluate the learned features, we use Zernike polynomials, as they are more discriminative compared to spherical harmonics (Ramasinghe et al., 2019a). We first train the network on the 55k ShapeNet objects by denoising spectral maps, and then apply the trained network on the ModelNet10 & 40. The features are then extracted from the bottleneck (similar to Sec. 5.3), and fed to a FC classifier (Table 4). We achieve state-of-the-art results in ModelNet40 with a simple pretext task.

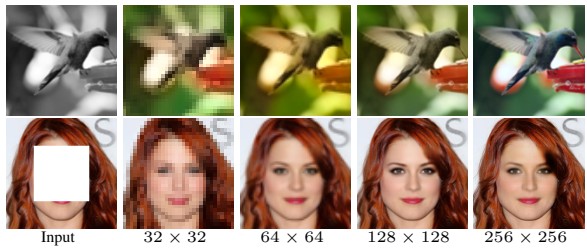

Input    32 × 32    64 × 64    128 × 128    256 × 256

Figure 19: Scalability: we subsequently add layers to the architecture to be trained on increasingly high-resolution inputs

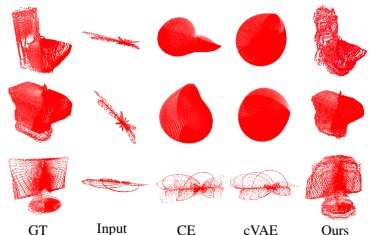

GT    Input    CE    cVAE    Ours

Figure 20: Qualitative comparison of 3D spectral denoising. The results are converted to the spatial domain for a clear visualization.

## 6 CONCLUSION

Conditional generation in multimodal domains is a challenging task due to its ill-posed nature. In this paper, we propose a novel generative framework that minimizes a family of cost functions during training. Further, it observes the convergence patterns of latent variables and applies this knowledge during inference to traverse to multiple output modes during inference. Despite using a simple and generic architecture, we show impressive results on a diverse set of tasks. The proposed approach demonstrates faster convergence, scalability, generalizability, diversity and superior representation learning capability for downstream tasks.

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
