# OpenReview forum: "Conditional Generative Modeling via Learning the Latent Space"
_ICLR.cc/2021/Conference — ICLR 2021 Poster_

### Official Review · AnonReviewer1 · 2020-10-28
**An interesting paper for conditional multimodal generation**

**Rating:** 7
**Confidence:** 3

**Review:**

The paper addressed the deterministic inference issue and enabled the conditional generation in multimodal spaces. The authors also explained the `` 'generalizability' advantage over cGANS, that is capable of learning more task-agnostic representations.
This paper is very dense. The authors conducted many experiments to show 1) the presented conditional generative modeling approach is capable of learning diverse representation and hence lead to diverse generation, 2) the proposed methods worked quite well across different tasks, both subjectively and objectively. Ablation studies are also performed to show ```'momentum' as a supplementary aid” is helpful.

At this moment, I don’t have major concerns regarding this submission. I incline to accept this paper, and am willing to further change my rating. Below are some minor comments:
1.	Based on the description in section 2 to 4, and also the strong experimental results, I am convinced that the proposed approach converges faster than cGAN, and can learn diverse representations and enables multimodal generation. However, it is a bit surprising/interesting that proposed method worked consistently better than other prior methods in almost all the downstream tasks explored. Can you elaborate more on why learned representations worked consistently well in almost all downstream tasks that have tried in the paper.
2.	The experimental study part is very dense. It would be good to have a short section clearly compare the model size/capacity of your models and the counterpart models. Also, is it good to move at least one model architecture you used to main text?
3.	The authors can consider comparing and contrasting with the normalizing flow related work. When normalizing flow applied for inference/learning latent representations, it seems (weakly) related to your design described on section 2.1

---

> ### Author Response · Authors · 2020-11-15
> **We added the capacity comparison and discussed the related work.**
>
> We appreciate the positive and informative comments by the reviewer.
>
> **Comment: Can you elaborate more on why learned representations worked consistently well in almost all downstream tasks that have tried in the paper**
>
> Although we currently do not have a rigorous proof, we would like to discuss some insights here.
>
> As explained by Remark 1, there is a general mismatch between the goals of $l_1$ loss and the adversarial loss. The $l_1$ loss assumes the distributions are Laplacian and tries to minimize the distance between them (without accounting for the variance), while the adversarial loss tries to minimize the JS divergence between the distributions. This difference in the optimization objectives causes the model to come to an equilibrium point, which is not perfectly stationed in the optimal data manifold. Therefore, it can be assumed that in a multimodal output space, for a given input, the extracted features by the generator are not optimal. In other words, if $l_1$ loss is dominant, the features represent the median of all the observed data, and if the adversarial loss is dominant, the representation can have a poor dependency on the specific output. We believe that this can be one of the reasons our model outperforms other models that use the adversarial loss.
>
> **Comment: The authors can consider comparing and contrasting with the normalizing flow related work.**
>
> Thank you for bringing our attention to this. We have now briefly mentioned Normalizing flows in Sec. 1 and a more detailed explanation (along with VAEs) in App. 1
>
>
> **Comment: It would be good to have a short section clearly compare the model size/capacity of your models and the counterpart models**
>
> We have now added this comparison to the paper. The #flops of the models are as follows for a batch size of 10.
>
> |   Model             |Flops    ($1 \times 10^{9}$)                      |
> |----------------|-------------------------------|
> |CE   |`0.634        |
> |PN           `|`0.946           |
> |Chroma          |1.275|
> |CIC          |52.839`|
> |P2P          | 0.723   `|
> |Izuka          |14.082`|
> |RFR          |25.64           `|
> |Ours          | `0.638  |
>
>
> **Comment: Also, is it good to move at least one model architecture you used to main text?**
>
> We have now moved model architecture for 128 input size to the main text.

---

### Official Review · AnonReviewer3 · 2020-10-28
**A new method for modeling continuous multimodal spaces**

**Rating:** 10
**Confidence:** 5

**Review:**

This paper proposes a family of cost functions and a framework for modeling a continuous multimodal (CMM) space.
The proposed model converges more stably and faster than conventional methods and shows high-quality results in several tasks. Also, this method can generate diverse outputs at inference with a single model. It was partially possible with many GAN methods, but there is a significant improvement in diversity.

- Clear motivation and well-defined method\
The problem of modeling the CMM space is well defined, and the limitation of the previous methods are described in detail. And the intuition to resolve this problem is also highly convincing.

- Various and extensive experiments\
The experimental settings and results support the effectiveness of the proposed method. The toy example in Figure 3 clearly shows that the proposed method can model CMM space properly.
Each experiment for downstream tasks also has a detailed explanation, showing good qualitative and quantitative performance.

There are no major weaknesses in the overall content. However, it is necessary to correct that table 5 and table 6 are overlapped, and many figures and tables are mixed in a somewhat disorderly manner.

It is somewhat similar to [1,2] in that a separate variable is defined for generating multiple outputs in a deterministic function. It's good to discuss this.

It would be great if there is a follow-up study to see if it could be extended to generate an image from random variables, like Conditional GAN. And some naming is required to represent the method.

[1] Auxiliary deep generative models, L. Maaløe, ,et al.
[2] Sym-parameterized Dynamic Inference for Mixed-Domain Image Translation, S. Chang, et al.

---

> ### Author Response · Authors · 2020-11-15
> **We have cited the mentioned papers**
>
> We highly appreciate the encouraging and valuable comments by the reviewer.
>
> **Comment: However, it is necessary to correct that table 5 and table 6 are overlapped, and many figures and tables are mixed in a somewhat disorderly manner.**
>
> We have now used extra space to make the presentation clearer. We hope the paper is now more reader-friendly.
>
> **Comment: It is somewhat similar to [1,2] in that a separate variable is defined for generating multiple outputs in a deterministic function. It's good to discuss this.**
>
> Thank you for mentioning these work, We have now cited the above work in Sec. 1 and added a more detailed explanation (along with VAEs and Normalizing flows) in App. 1
>
> **Comment: It would be great if there is a follow-up study to see if it could be extended to generate an image from random variables, like Conditional GAN.**
>
> Thank you for the valuable pointer. We are already working on a similar followup study, where we traverse through the latent space to generate diverse data using a conditional GAN. In this work, we have already removed the dependency of the generator on the  separate z predictor network by applying a bijective mapping between the latent space and the output manifold (without using a cyclic consistency loss).
>
> **Comment:  some naming is required to represent the method.**
>
> We have mentioned the name "Conditional Generation by Modeling the Latent Space (cGML)" for our model in the paper.

---

### Official Review · AnonReviewer4 · 2020-10-28
**Good Model, the Paper Can Be Improved**

**Rating:** 6
**Confidence:** 3

**Review:**

Summary:
In this paper, the authors proposed a general-purpose framework for conditional generation in multimodal space. The proposed method is optimized to find multi-modal optimal solutions at inference time. This general method can be applied to a lot of down-stream tasks, improving inference performance without the need to carefully design a network structure.

Pros:
1. The proposed method is very general. It is evaluated on quite a few different tasks, including sketch to image generation, image imputation. image colorization, etc.
2. The proposed method shows superior results both qualitatively and quantitatively compared with baseline models. It is quite impressive how comprehensive the evaluation is. The multi-modality perspective is especially well presented.

Cons:
1. The authors should carefully audit the paper. There are quite a few typos and presentation flaws, which makes it much harder to read. For example, the numbering of the figures is not consistent. There is not figure 2, 4, 6, 8 etc.
2. I understand that the authors want to show as many results as possible, but the presentation becomes very crowded, especially on page 8. I would recommend the authors to leave the critical experimental results here and move some to the supplementary materials.
3. Despite the impressive experimental results, I think readers would benefit more from a deep discussion. For example, what is the relationship between this model with a VAE model? The similarity is of course the continuous latent space. By varying the latent factor, VAE models can generate diverse data as well.

---

> ### Author Response · Authors · 2020-11-15
> **We  have added more discussions.**
>
> We are thankful to the reviewers for the positive and valuable comments.
>
> **Comment: There are quite a few typos and presentation flaws, which makes it much harder to read. For example, the numbering of the figures is not consistent.**
>
> Thank you for mentioning this. We have now corrected this.
>
>
> **Comment: Presentation becomes very crowded,  especially on page 8.**
>
> We have now used extra space to reduce the clutter. We hope now the presentation is clearer.
>
>
> **Comment: what is the relationship between this model with a VAE model? The similarity is of course the continuous latent space. By varying the latent factor, VAE models can generate diverse data as well.**
>
> Thank you for drawing our attention to this. A detailed comparison summary along with VAE, Normalizing flows and some other related works is now added to App. 1.
>
> To summarise, in VAEs, the posterior distribution is typically sampled from the family of Gaussians. However, assuming the posterior distribution of the latent space as a Gaussian distribution can constrain the quality of the generated data distribution, as the true distribution may be far from a Gaussian (note that recent work such as [1] and [2] have been proposed to address this).
>
> In contrast, we do not explicitly model our latent space as a probability distribution. However, we can draw some interesting analogies from a probabilistic perspective as follows: our latent space $\zeta$ can be interpreted as a set of energy surfaces $E_{x_j}:\zeta \rightarrow \mathbb{R}$, as $E_{x_j} = ||{y^g_{j} - G(x,z_{j})}||$ for each ground truth mode $y^g_{j}$. From this perspective, Fig.~21 in the appendix illustrates the energy heatmaps for the toy example. As shown,  high energies are indicated by a brighter color. Since our system has a finite energy, the combined energy $E_x = \sum_j E_{x_j}$ can be transformed to a probability distribution via the Gibbs measure as $p'(z) = \frac{1}{T(\beta)} \exp (-\beta E_x(z))$, where $T(\cdot)$ is the partition function.This probability is not restricted to a Gaussian, as opposed to VAEs.
>
> Another critical difference between the VAEs and our model is that we do not sample directly from $p'(z)$, since to obtain $p'(z)$, we need to integrate $E_x$ over the latent space. However, our predictor network $\mathcal{Z}$ learns the high probability coordinates $\{z^*}$ of $p'(z)$, and is able to converge to such locations at inference. This probabilistic perspective of our latent space  is intuitively justified by the convergence samples shown in Fig. 41 in appendix. The intermediate samples we obtain as we go from $z$ to $z^*$ also produce plausible results, however, the visual quality at the $z^*$ is maximized, indicating high $p'(z =z^*|x)$. Therefore, our model does not explicitly learns the probability distributions, rather the  predictor network learns to converge to the high probability areas in complex distributions.
>
>
> [1] - Maaløe, Lars, et al. "Auxiliary deep generative models." arXiv preprint arXiv:1602.05473 (2016).
> [2] - Rezende, Danilo Jimenez, and Shakir Mohamed. "Variational inference with normalizing flows." arXiv preprint arXiv:1505.05770 (2015).

---

### Official Review · AnonReviewer2 · 2020-10-30
**Conditional Generative Modeling via Learning the Latent Space**

**Rating:** 7
**Confidence:** 3

**Review:**

Quality：
The proposed general-purpose framework for modeling CMM spaces is worthwhile and insightful. By using a set of domain-agnostic regression cost functions instead of the adversarial loss, it improves both the stability and eliminates the incompatibility between the adversarial and reconstruction losses, allowing more precise outputs while maintaining diversity.

However, it would be interesting to see the qualitative and quantitative comparison with the latest related works. For example, the following two CVPR2020 papers(For reference only):
[1] Zheng C, Cham T J, Cai J. Pluralistic image completion[C]//Proceedings of the IEEE Conference on Computer Vision and Pattern Recognition. 2019: 1438-1447.
[2] Li J, Wang N, Zhang L, et al. Recurrent Feature Reasoning for Image Inpainting[C]//Proceedings of the IEEE/CVF Conference on Computer Vision and Pattern Recognition. 2020: 7760-7768.

Clarity：
The paper is very well-written and well-structured and is friendly for readers to understand. It starts off by pointing out the key shortcomings of the use of a combination of reconstruction and adversarial losses and then clarifies the framework proposed in this paper for modeling CMM spaces. Continuously, it explains the drawbacks of conditional GAN methods and illustrates the idea via a toy example. After that, with an extensive set of experiments, this paper experiment on several such tasks with different datasets. It illustrates the outperformance in both qualitative and quantitative experimental comparison and demonstrates state-of-the-art performance.

By the way, it may be more friendly for readers to read the paper if the layout of diagrams and tables can be reformatted in a clearer way.

Originality and Significance：
Due to the ill-posed nature of conditional generation in multimodal domains, this paper proposed a novel generative framework with a simple and generic architecture instead of the adversarial loss. The proposed approach demonstrates faster convergence, scalability, generalizability, diversity, and superior representation learning capability for downstream tasks. At the same time, the comparable performance has been validated on different datasets both quantitatively and qualitatively. And in most of the experiments, it achieves state-of-the-art performance.

Based on the above considerations, I think it is a good paper that marginally above the acceptance threshold. And it may be worthwhile to be accepted if more latest experimental comparison can be shown, and still have the outperformance.

-----------------------------------------------------------------------------------------------
Compared with the latest related work, the author added the qualitative and quantitative comparison against "Recurrent Feature Reasoning for Image Inpainting" in the image completion task. And it shows the outstanding performance of this paper.
They have also reformatted the paper so the paper becomes clearer for readers to read.
Consider the above all, I think it’s a good paper and is worthwhile to be accepted.


So I improving my rating to “7: Good paper, accept”

---

> ### Author Response · Authors · 2020-11-15
> **We have added more comparisons**
>
> We thank the reviewers for the positive and insightful comments.
>
> **Comment: "It would be interesting to see the qualitative and quantitative comparison with the latest related works."**
>
> Reply: We added the comparison against "Recurrent Feature Reasoning for Image Inpainting" in the image completion task. We trained their model under the same settings provided to other models for 150 epochs and have added both qualitative and quantitative results to Fig. 9  and Table 3, respectively. The results are as follows:
>
> 10% corruption
>
> |LPIP     ` |PieAPP    ` |PSNR      `        |  SSIM                       `  |
> |----------------|-------------------------------|-----------------------------|-----------------------------|
> |0.05|`0.74       | 29.31 |0.85     |
>
>
>
> 15% corruption
>
> |LPIP     ` |PieAPP    ` |PSNR      `        |  SSIM                       `  |
> |----------------|-------------------------------|-----------------------------|-----------------------------|
> |0.09|`1.03       | 19.22 |0.70     |
>
>
>
> 25% corruption
>
> |LPIP     ` |PieAPP    ` |PSNR      `        |  SSIM                       `  |
> |----------------|-------------------------------|-----------------------------|-----------------------------|
> |0.17|`1.13       | 18.42 |0.66     |
>
>
>
>
>
> **Comment: "It may be more friendly for readers to read the paper if the layout of diagrams and tables can be reformatted in a clearer way."**
>
> Thank you for pointing this out. We have reformatted the paper in order to reduce the clutter. We hope it is more friendly to the readers now.

---

### Author Response · Authors · 2020-11-12
**Common response to all the reviewers.**

We thank all the reviewers for their positive and insightful comments. We are currently working on improving our paper based on them.

We will individually address each of the issues mentioned and upload a revised draft soon. We will also reply to each reviewer highlighting the modifications and discuss the insightful points raised by the reviewers.

We highly appreciate the effort and time devoted by reviewers while evaluating our paper.

---

### Decision · Program_Chairs · 2021-01-07
**Final Decision**

**Decision:**

Accept (Poster)

**Comment:**

The paper proposes a model and a training mechanism for multimodal generation. The reviews are generally positive: they praise the generality of the method, the extensive experimental evaluation, and the good empirical results. Overall, no major concerns were raised, and all reviewers recommend acceptance.

A couple of concerns remain, in my view:
- The method is generally heuristic, and intuitively rather than theoretically motivated. This is compensated of course by the empirical evaluation, which is thorough.
- The paper could be better written. The reviewers suggested some minor improvements which were implemented in the updated version, but I believe there is room for further improvement.

Due to the above concerns, I consider the rating of reviewer #3 (10: Top 5% of accepted papers, seminal paper) to be unjustifiably high. On balance, however, I'm happy to recommend acceptance.

Message to the authors:

In the abstract you write: "a simple generic model that can beat highly engineered pipelines". Please be aware that the word "beat" evokes competition, winners and losers, so it's not appropriate in the context of scientific evaluation. Please consider replacing it with something neutral, such as "a simple generic model that can perform better than ...".